# ZEROS CAN BE INFORMATIVE: MASKED BINARY U-NET FOR IMAGE SEGMENTATION ON TENSOR CORES

**Chunshu Wu**[1], **Ruibing Song**[2], **Sushant Kondguli**[3], **Tong Geng**[2], **Ang Li**[1,4]
Pacific Northwest National Laboratory[1], Rice University[2], Meta[3], University of Washington[4]

## ABSTRACT

Real-time image segmentation is a key enabler for AR/VR, robotics, drones, and autonomous systems, where tight accuracy, latency, and energy budgets must be met on resource-constrained edge devices. While U-Net offers a favorable balance of accuracy and efficiency compared to large transformer-based models, achieving real-time performance on high-resolution input remains challenging due to compute, memory, and power limits. Extreme quantization, particularly binary networks, is appealing for its hardware-friendly operations. However, two obstacles limit practicality: (1) severe accuracy degradation, and (2) a lack of end-to-end implementations that deliver efficiency on general-purpose GPUs.

We make two empirical observations. (1) An explicit zero state is essential: training with zero masking to binary U-Net weights yields noticeable sparsity. (2) Quantization sensitivity is relatively uniform across layers. Motivated by these findings, we introduce Masked Binary U-Net (MBU-Net), obtained through a cost-aware masking strategy that prioritizes masking where it yields the highest accuracy-per-cost, reconciling accuracy with near-binary efficiency.

To realize these gains in practice, we develop a GPU execution framework that maps MBU-Net to Tensor Cores via a subtractive bit-encoding scheme, efficiently implementing masked binary weights with binary activations. This design leverages native binary Tensor Core BMMA instructions, enabling high throughput and energy savings on widely available GPUs. Across 3 segmentation benchmarks, MBU-Net attains near full-precision accuracy (3% average drop) while delivering $2.04\times$ speedup and $3.54\times$ energy reductions over a 16-bit floating point U-Net. The code is available at https://github.com/ChunshuWu/MBU-Net.

## 1 INTRODUCTION

Image segmentation is essential for applications such as AR/VR, autonomous drones, autopilot systems, and robotics. In these domains, real-time perception and sustained operation are typically required. For instance, segmentation at roughly 60 Hz frame rate is fundamental for Meta's AR glasses to deliver a smooth, context-aware user experience. Likewise, inspection or delivery drones often must operate continuously for 30 minutes, sometimes even hours on a single battery. Consequently, in addition to accuracy, processing speed and power efficiency are also decisive for practical deployment on edge devices. With rising demand for high-resolution data and high frame rates, these constraints have become increasingly stringent.

Compared to large, over-parameterized architectures such as Vision Transformer (Dosovitskiy et al., 2020), U-Net (Ronneberger et al., 2015) is substantially cheaper in compute, memory, and energy for dense prediction, making it a more promising choice for edge scenarios. Additionally, U-Net's characteristic encoder-decoder structure with skip connections has demonstrated high effectiveness for pixel-level tasks (Chen et al., 2018; Zhou et al., 2019), preserving spatial details and facilitating accurate pixel-level predictions. Nevertheless, running a U-Net on high-resolution video in real time can still exceed the strict power, latency, and memory limits of many edge devices – further optimizations are necessary to reconcile accuracy with deployability.

Quantization (Gholami et al., 2022; Wei et al., 2024; Haghi et al., 2024) is a natural path to efficiency. At the extreme, binary networks (Hubara et al., 2016; Liu et al., 2020) use 1-bit weights and activations, replacing Multiply-Accumulations (MACs) with simple bitwise operations (XNOR/XOR)

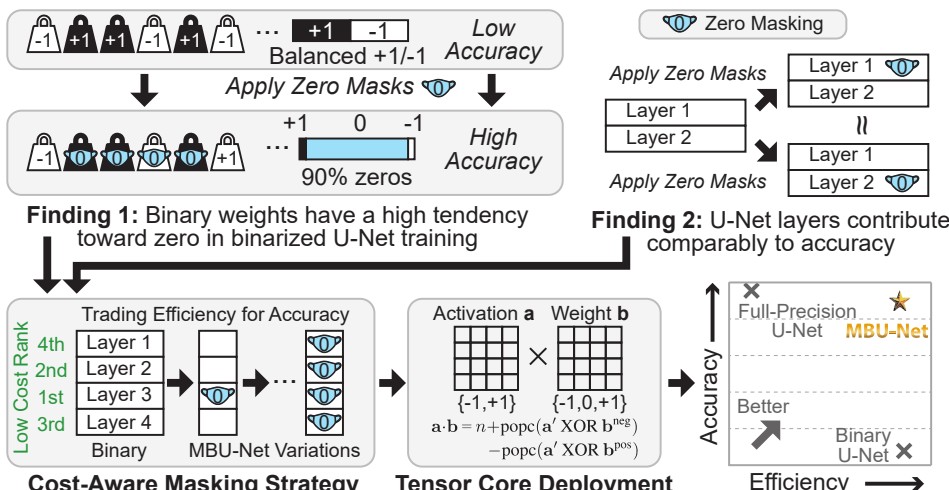

Figure 1: Overview. Obtained through a cost-aware masking strategy and optimized for Tensor Core on GPUs, MBU-Net achieves full-precision level accuracy and binary-level efficiency.

followed by a population count (popcount) (Rastegari et al., 2016). Since most processors handle data in chunks rather than individual bits, practical implementations group many binary values together – allowing a single instruction to execute many binary operations in parallel. This approach is highly efficient on CPUs and GPU CUDA cores (Li et al., 2019; Chen et al., 2023), and can be further extended to exploit Tensor Cores (Li & Su, 2020). Collectively, these techniques leverage inexpensive, hardware-friendly operations to sharply reduce compute, memory traffic, and energy, making extremely low-bit inference attractive for resource-constrained edge deployments.

However, two practical challenges hinder the effectiveness of extreme quantization. (1) **Accuracy degradation**. Binary networks usually suffer from a significant drop in accuracy due to the aggressive 1-bit quantization, and the impact on U-Net in segmentation tasks can be especially severe (AskariHemmat et al., 2019; Guo et al., 2022). (2) **Scarce practical solutions on general-purpose processors**. Many binary/ternary methods (Wan et al., 2018; Heinrich et al., 2018; Zhu et al., 2022; Yin et al., 2024) are algorithmic proofs-of-concept or tailored to custom FPGA (Liang et al., 2018; Geng et al., 2019; Mani et al., 2022) and ASIC (Wagle et al., 2020) datapaths, leaving end-to-end U-Net designs and optimizations on general-purpose processors (e.g., GPUs) underexplored. To address these challenges, this paper aims to answer: (1) How to achieve near-full-precision accuracy with near-binary efficiency for U-Net on segmentation tasks? (2) How to deliver that efficiency on widely available GPUs without relying on specialized accelerators?

To address the first challenge, we design our solution based on two key observations: (1) **An explicit zero state is essential.** Pure binary representations ({-1, +1}) force every connection to contribute, offering no neutral state to suppress uncertain or noisy signals. Our experiments demonstrate that, when training with zero masking on U-Net weights, in addition to substantially enhanced accuracy, there is noticeable sparsity – over 90% zeros in many layers, some consistently exceeding 95% – far outnumbering the +1 and -1 states. This indicates an abundance of signals that should be suppressed by zeros. (2) **Uniform sensitivity across layers.** An exhaustive sweep over 4,000 per-layer quantization configurations shows that masking any single layer has a comparable effect on accuracy. Since masking adds overhead, and masking all layers is often unnecessary if a subset already provides sufficient suppression of information flow, low-cost layers can be prioritized for masking to balance computation and accuracy. For example, masking transposed convolution layers has a strong impact on accuracy while incurring negligible computational cost.

Since extra costs are incurred due to masking, we can strategically pick which layer to enable masking for computing and accuracy trade-off, as it may not be necessary to apply masking for all layers, some of them can already ensure sufficient suppression for information pass. For instance, transposed convolution layers have strong impact on accuracy, despite their negligible computing cost. Guided by these findings, we propose a *cost-aware masking* strategy that identifies critical layers where masking is most beneficial, and introduce *Masked Binary U-Net* (MBU-Net), which achieves **near full-precision accuracy** with **near-binary efficiency** as highlighted in Figure 1.

To address the second challenge, we target Tensor Cores, mapping MBU-Net via bit-packing and dedicated APIs. On GPUs with Tensor Cores, performance is typically memory-bound rather than compute-bound, thus we retain binary activations to reduce data movement and power. This approach is timely: Tensor Cores are now common even on edge GPUs (e.g., NVIDIA Jetson Xavier with Volta Tensor Cores, Jetson Orin with Ampere Tensor Cores), positioning them as practical platforms for efficient edge application deployment. Although prior frameworks have accelerated pure Binary Neural Networks (BNNs) on GPUs (Li et al., 2019; Li & Su, 2020), to the best of our knowledge, no high-performance framework exists for selectively masked binary models on Tensor Cores, especially for U-Net. To bridge this gap, we introduce a *subtractive bit-encoding* method that naturally realizes masked binary weights with binary activations within the Tensor Core paradigm, thereby unlocking **substantial efficiency gains on modern commodity GPUs**. Our major contributions are listed below.

1. We discover that: (1) An explicit zero masking state in the weights of a binary U-Net can significantly improve the segmentation quality, closely approaching full-precision results. (2) Despite being cheap, transposed convolution layers are accuracy-critical.

2. We employ a cost-aware masking strategy based on above discoveries, systematically identifying the most critical layers in a U-Net, leading to MBU-Net models that regain near full-precision segmentation accuracy at minor masking cost over a pure BNN.

3. We design and implement an end-to-end acceleration framework with native BMMA binary Tensor Core instructions through subtractive bit-encoding method, enabling unified and high-throughput inference for MBU-Net models on GPU Tensor Cores.

4. Our approach delivers an average $2.04\times$ speedup and $3.54\times$ power efficiency improvement over a 16-bit floating point U-Net on A100, H100, Jetson Orin Nano, and RTX 2080 Ti GPUs, while incurring only 3% average accuracy loss on 3 segmentation datasets. The U-Net contains 16M parameters, and the input images range from 0.25M to 0.6M pixels.

## 2 BACKGROUNDS

### 2.1 U-NET

Since its introduction in 2015 (Ronneberger et al., 2015), the U-Net architecture has proven to be far more than just a successful model for biomedical image segmentation (Ibtehaz & Rahman, 2020; Cheng et al., 2022). It has been extended to other domains including image enhancement (Komatsu & Gonsalves, 2020), generative AI (Ho et al., 2020; Schonfeld et al., 2020; Si et al., 2024), scientific modeling (Kamali & Laksari, 2024; Zhu et al., 2025), etc.

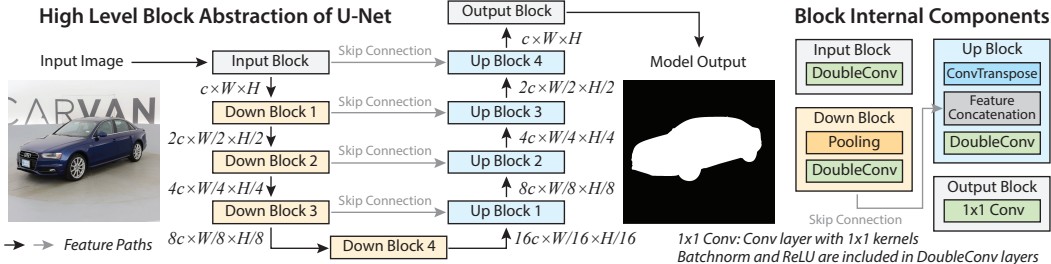

Figure 2: Left: Conventional U-Net architecture exemplified with segmentation task. $c$: number of channels, $W/H$: feature width / height. Right: internal components of each abstracted block.

At its core, as Figure 2 illustrates, U-Net is a symmetric encoder-decoder architecture linked by skip connections. The encoder (contracting path with orange down blocks) follows the typical structure of a convolutional network, downsampling the input to capture semantic context. The decoder (expanding path with blue up blocks), in return, progressively upsamples and refines the feature representations to construct a full-resolution image. To avoid forcing all information through a narrow bottleneck, skip connections concatenate encoder features with the corresponding decoder features after transposed convolution layers, passing information along multiple paths and elegantly preserving information at different semantic hierarchies.

This foundational design has inspired a wide range of successors. For instance, U-Net++ (Zhou et al., 2019) introduces nested and dense skip connections, while UNet3+ (Huang et al., 2020) extends this with full-scale skip connections. Other variants integrate mechanisms such as attention gates (Oktay et al., 2018) or residual connections (Diakogiannis et al., 2020; Alom et al., 2019) to enhance capacity. More recently, Transformer-based approaches (Dosovitskiy et al., 2020; Strudel et al., 2021) and Segment Anything Models (SAM) (Kirillov et al., 2023; Ravi et al., 2024) have achieved exceptional segmentation quality. However, the massive parameter counts and computational demands of these large models pose significant challenges for edge deployment.

In fact, even for the standard U-Net, the computational footprint at real-time, high-resolution settings is substantial. A configuration at 720p can require on the order of hundreds of GFLOPs per frame, and sustaining 60 fps implies tens of TFLOPs per second plus heavy memory bandwidth from skip connections. On edge platforms such as AR glasses and drones, on-device inference is often necessary for latency, privacy, and reliability, yet power budgets are only a few watts. These constraints motivate aggressive quantization and sparsity to reduce arithmetic and bandwidth while preserving accuracy. We therefore investigate U-Net in the binarized regime, aiming to characterize its behavior under extreme precision reduction and masking, and to provide guidance for edge-oriented designs that balance accuracy, compute, and energy.

In addition to segmentation, U-Net has been widely explored for image enhancement tasks, including denoising (Fan et al., 2022; Tripathi, 2021) and super-resolution (Hu et al., 2019). With the rapid growth of AR/VR, U-Net has emerged as a promising candidate for waveguide correction (Chapiro et al., 2024) to enable lower power consumption and reduced distortions in AR systems. Moreover, U-Net plays a pivotal role in generative AI, serving as the backbone for models such as DDPM (Ho et al., 2020) and high-resolution image synthesis (Rombach et al., 2022).

## 2.2 Binary Neural Networks

The success of Deep Neural Networks (DNNs) has been largely predicated on a trend of escalating model complexity. In contrast, BNNs (Hubara et al., 2016) reduce the complexity to an extreme extent, with both activations and weights constrained to binary values, typically represented as $\{-1, +1\}$. For activations, this constraint is commonly enforced through a binarization function $x_b = \text{sign}(x)$ during forward pass, whereas the weights are trained under binary representation, typically using Straight-Through Estimators (STE) (Bengio et al., 2013; Courbariaux et al., 2015; Yin et al., 2019) to approximate the gradient in such non-differentiable architecture. Despite the low resolution, theoretical analysis (Anderson & Berg, 2017) has confirmed BNN's capability of capturing features.

For efficient practical deployment, XNOR-Net (Rastegari et al., 2016) was introduced. Specifically, for binary vectors $\mathbf{a}|a_i \in \{-1, +1\}$ and $\mathbf{b}|b_i \in \{-1, +1\}$, their MAC operation can be translated into bit-wise operations and population count on bit vectors $\mathbf{a}'|a_i' \in \{0, 1\}$ and $\mathbf{b}'|b_i' \in \{0, 1\}$:

$$\mathbf{a} \cdot \mathbf{b} = \sum_i^n (2a_i' - 1)(2b_i' - 1) = 2 \cdot \text{popc}(\mathbf{a}' \text{ XNOR } \mathbf{b}') - n \tag{1}$$

using the equality $a_i' \text{ XNOR } b_i' = 2a_i'b_i' - a_i' - b_i'$. Here, "popc" refers to population count operation that counts the number of 1's in an $n$-bit vector, e.g., a 32-bit vector as an unsigned integer.

In the past decade, BNN research follows two main threads. On the algorithmic level, researchers strive to improve BNN training accuracy, while on the implementation level, the main target is to minimize power and latency, especially on FPGA and ASIC. In this work, our goal is to push the boundaries on both directions for U-Net, achieving near full-precision accuracy with near-binary efficiency.

## 2.3 Ternary Neural Networks

Table 1 summarizes a few notable studies related to neural networks with ternary activations or weights. TWN (Li et al., 2016) and TTQ (Zhu et al., 2016) focus on full-precision activation and ternary weights on the algorithmic level, reporting results primarily on image classification tasks. TBN (Wan et al., 2018) adopts ternary activations and binary weights, also at the algorithmic level.

Table 1: Summary of ternary activation/weight network studies.

| Related Work | TWN | TTQ | TBN | TernaryNet | FATNN |
|---|---|---|---|---|---|
| Platform | Unspecified | Unspecified | Unspecified | CPU | GPU |
| Focus | Acc[*] | Acc | Acc & Theoretical Speed | Acc | Acc & Speed |
| Activation | Full-Precision | Full-Precision | Ternary | Ternary | Ternary |
| Weights | Ternary | Ternary | Binary | Ternary | Ternary |
| Task | IC[*] | IC | IC | Segmentation | IC |

[*] IC: Image Classification; Acc: Accuracy

TernaryNet (Heinrich et al., 2018) is also an algorithmic endeavor, focusing on achieving faster inference without GPUs. FATNN (Chen et al., 2021) introduces a 2-bit encoding method: $-1 \rightarrow 00$; $0 \rightarrow 01/10$, $+1 \rightarrow 11$ and uses a series of bit operations to reduce ternary-ternary operation complexity from $O(4N)$ to $O(2N)$. TAB (Zhu et al., 2022) proposes a flexible framework accommodating various activation/weight quantization schemes. To our knowledge, none of these works analyze per-layer contributions to accuracy, assess compatibility with U-Net, or deliver end-to-end implementations targeting Tensor Cores on commodity GPUs.

## 2.4 TENSOR CORE APIS

Tensor Cores are specialized units that accelerate Matrix Multiply-Accumulate (MMA), computing $D = A \times B + C$ per instruction, where $A$, $B$, $C$, and $D$ are matrices of compatible dimensions. Compared with issuing many MACs on CUDA cores, a single MMA instruction performs a large number of fused operations on a small matrix tile. For example, the throughput of Tensor Cores on A100 is almost an order of magnitude higher than CUDA cores, emphasizing the trend of shifting AI workloads to such dedicated matrix engines for higher throughput and efficiency.

In GPUs, processing units are grouped into warps, and the above MMA is executed using the WMMA API with the following three key functions: (1) Load data from memory to registers. (2) Perform the above MMA operation. (3) Write the data back to memory. Less widely known, some NVIDIA GPUs also expose bit MMA (namely, BMMA): Tensor Core instructions that compute binary matrix products using bitwise logic (e.g., XOR/AND) with popcount accumulation. These binary Tensor Core features remain low-level and experimental, as they are not exposed by mainstream libraries (e.g., cuBLAS, cuDNN) and typically require custom kernels. Consequently, despite their potential, binary Tensor Core capabilities remain underused and underexplored in practice.

In this work, we extend the scope of an existing BNN acceleration framework (Hosseini et al., 2019; Li & Su, 2020) on Tensor Core, bringing the unique BMMA capability under the spotlight.

## 3 METHODS

In this section, we present MBU-Net, a class of masked binary U-Nets inspired by empirical observations (§3.1.1) and constructed using a cost-aware masking strategy in §3.1.2, achieving balanced trade-off between accuracy and computational cost. Next, we demonstrate that with subtractive bit encoding §3.2.1, masked layers can be naturally mapped to Tensor Cores by reusing the BNN mapping §3.2.2, enabling an end-to-end implementation that runs efficiently on modern edge GPUs.

### 3.1 MASKED BINARY U-NET

#### 3.1.1 EMPIRICAL OBSERVATIONS AND ANALYSIS

We begin by defining a U-Net backbone with 12 abstracted configurable layers: $4\times$ double convolution layers in the encoder, labeled as down-C1$\sim$4; $4\times$ transposed convolution layers in the decoder, as up-CT1$\sim$4; $4\times$ double convolution layers in the decoder, as up-T1$\sim$4. The indexing is consistent with the block indices in Figure 2. Each layer can be instantiated in one of two states: binary or masked (i.e., with an explicit zero state in the weights), resulting in a $2^{12}$ design space. In the exhaustive sweeping, Carvana dataset (Shaler et al., 2017) is used as a representative example.

In binary layers, input features, output features, and weights are all binary values within $\{-1, +1\}$, following the convention of BNNs. In masked layers, a masking state '0' is included in weights,

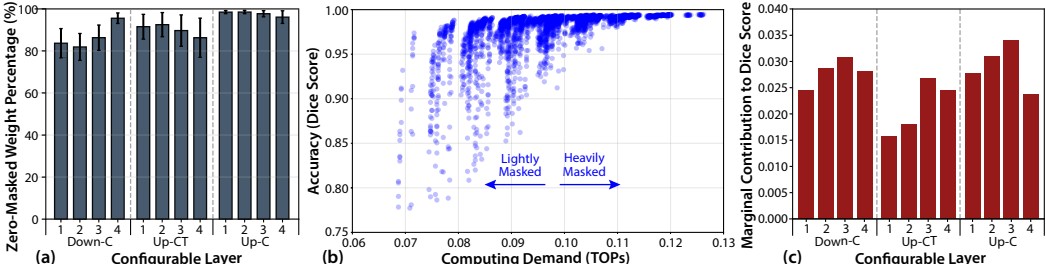

Figure 3: (a) Statistics of the weights masked to zero. (b) Accuracy and computing demand distribution for all layer configurations. (c) Shapley analysis: marginal improvement in Dice score from masking each binary layer. Only cases with fewer than five masked layers are included for clarity.

effectively making the weights ternary $\{-1, 0, +1\}$, while keeping input and output features binary. Detailed network architecture is in Appendix A.2. Extensive investigations (Appendix A.3) demonstrate that higher-bit weight quantizations result only in marginal improvement compared to binary weights with masks. Through an STE-based training approach, the zero masking is automatically applied. The training configuration is in Appendix A.4.

Figure 3(a) demonstrates that zero weights are notably prevalent across all masked layers and configurations. On average, each layer exhibits more than 80% sparsity, with many layers exceeding 90% and some consistently surpassing 95%. This finding indicates that it is preferable to mask out the majority of weights while retaining only a small essential subset. Figure 3(b) supports this observation: layer configurations with heavier masking generally achieve higher Dice scores (from 0 to 1, higher is better), confirming the value of masked binary weights.

However, masking inevitably introduces additional costs, which can sometimes be heavy in computation. Figure 3(b) further shows that fully masking all layers can require about twice as many operations as lightly masked configurations. Meanwhile, a configuration requiring 0.08 TOPs can achieve accuracy comparable with the fully masked configuration exceeding 0.12 TOPs. This indicates that masking every layer is unnecessary when a small set of critical layers suffices, motivating us to identify those layers to achieve high accuracy at low cost.

Figure 3(c) shows the marginal gain in Dice score for each layer under zero masking. For clarity, we restrict the analysis to cases with fewer than 5 masked layers, since including more layers dilutes the per-layer contribution due to the limited improvement in Dice score. Interestingly, the contributions are broadly comparable across layers. This observation brings a straightforward yet practical strategy: prioritize masking low-cost layers.

### 3.1.2 COST-AWARE MASKING STRATEGY

The above empirical findings indicate that if some layers are significantly cheaper than others, masking the cheap layers can achieve substantial improvement in accuracy at minimal cost. We thus further profile the per-layer computing and memory in Figure 4, where the operations account for both multiplications and additions.

We observe that, the numbers of operations of transposed convolution layers (highlighted with red dashed frame) in the upsampling path are 1~2 orders of magnitude fewer compared to other layers, and the operation count and parameter count are relatively imbalanced across layers. As it is usually infeasible to perform analysis like Figure 3 across the entire design space, it is preferable to set up the strategy based on the model characteristics in Figure 4. Particularly, we define the cost-aware strategy using the weighted cost score for layer $l$:

$$s_{\text{cost}}^l = w_{\text{op}}\hat{n}_{\text{op}}^l + w_{\text{param}}\hat{n}_{\text{param}}^l \tag{2}$$

where $\hat{n}_{\text{op}}^l$ and $\hat{n}_{\text{param}}^l$ are the number of operations and the number of parameters of layer $l$ normalized to $[0, 1]$, and the weights $w_{\text{op}}$ and $w_{\text{param}}$ satisfy $w_{\text{op}} + w_{\text{param}} = 1$, as hyperparameters. The weighted cost scores are then ranked from low to high as the priority list for masking. Following the strategy, MBU-Net can be populated based on the priority list based on demand, narrowing down the design space significantly.

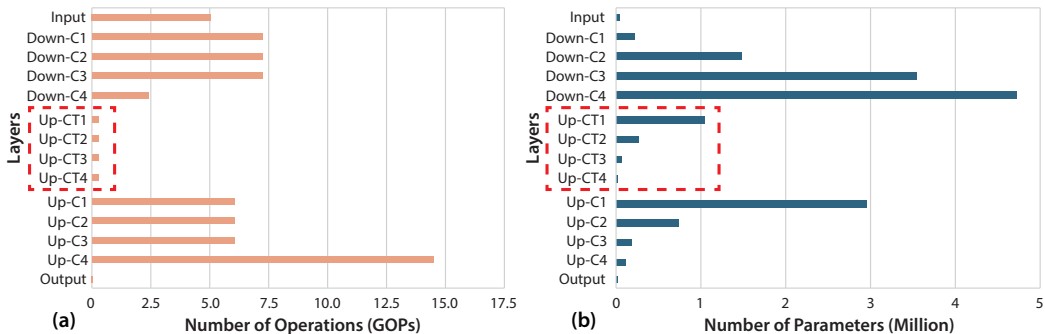

Figure 4: Per-layer cost profiling. (a) Number of addition and multiplication operations in each layer with a 512×512 image as a representative example. (b) Number of parameters in each layer.

## 3.2 TENSOR CORE DEPLOYMENT

### 3.2.1 SUBTRACTIVE BIT-ENCODING

MBU-Net maintains binary activations and assigns each layer to either binary or masked weights according to the cost-aware strategy described above. To ensure that the MAC between binary activation and masked weight can be naturally supported by native operations on Tensor Cores – XOR and popcount, we encode masked weights with two subtracting bit-planes. In particular, let $\mathbf{a}|a_i \in \{-1, +1\}$ denote a binary activation vector, and $\mathbf{b}|b_i \in \{-1, 0, +1\}$ as a masked weight vector, we encode a weight value as $b_i = b_i^{\text{pos}} - b_i^{\text{neg}}$, with $b_i^{\text{pos}}, b_i^{\text{neg}} \in \{0, 1\}$. The MAC operation is thus translated using $a_i' \in \{0, 1\}$:

$$\mathbf{a} \cdot \mathbf{b} = \sum_i^n \underbrace{2a_i'b_i^{\text{pos}} - b_i^{\text{pos}} - a_i'}_{1 - (a_i' \text{ XOR } b_i^{\text{pos}})} + \underbrace{a_i' + b_i^{\text{neg}} - 2a_i'b_i^{\text{neg}}}_{a_i' \text{ XOR } b_i^{\text{neg}}} = n + \text{popc}(\mathbf{a}' \text{ XOR } \mathbf{b}^{\text{neg}}) - \text{popc}(\mathbf{a}' \text{ XOR } \mathbf{b}^{\text{pos}})$$

(3)

Compared to Equation 1, the binary-ternary MAC operation is fully represented by bit-friendly operations, only to use XOR instead of XNOR to be compatible with Tensor Core intrinsics.

### 3.2.2 MAPPING LAYERS TO TENSOR CORE

We map these computations to GPU Tensor Cores using binary WMMA, or the experimental BMMA API. Each kernel of a masked layer stores the two bit planes as bit-packed matrices. At warp level, we operate on $8 \times 8 \times 128$ bit tiles, where the three dimensions are mapped to batch size, output layer number, and input layer number. For clarity, we show the high-level procedure for convolution on Tensor Core in Algorithm 3.2.2.

---

**Algorithm 1** Bit Convolution with Subtractive Bit-Encoding on Tensor Core

---

1: **for all** $H, W, C_{\text{out}}, B$ **do**                      ▷ feature height, width, output channel, batch size
2:     Initialize accumulators $C^{\text{pos}}, C^{\text{neg}} \leftarrow 0$
3:     **for all** $F_H, F_W$, and $C_{\text{in}}$ of 128-bit blocks **do**            ▷ filter height, width, input channel
4:         Load binary activation fragment $A$                                    ▷ row-major
5:         Load binary weight fragments $B_{\text{pos}}, B_{\text{neg}}$                          ▷ column-major
6:         bmma_sync($C^{\text{pos}}, A, B^{\text{pos}}, C^{\text{pos}}$)                          ▷ binary WMMA API
7:         bmma_sync($C^{\text{neg}}, A, B^{\text{neg}}, C^{\text{neg}}$)
8:     **end for**
9:     $R \leftarrow C^{\text{neg}} - C^{\text{pos}}$                                      ▷ temporary result $R$
10:     Threshold comparison: $A_{\text{out}} \leftarrow (R \geq \theta)$  ▷ covering batchnorm, bias, and binary activation
11:     Pack bits via intrinsics (ballot, brev) and store to memory
12: **end for**

---

In the algorithm, the matrices $A$, $B$, and $C$ follow the update rule $C \leftarrow A \cdot B + C$, implemented using the Tensor Core API function "bmma_sync". The function is further compiled to a Parallel Thread Execution (PTX) Tensor Core opcode that includes XOR and popcount. $\theta$ denotes the pre-computed threshold. The threshold comparison jointly incorporates batch normalization (Ioffe &

Szegedy, 2015), bias, and binary activation, which is a standard practice in BNNs. In terms of other layers (detailed in Figure 2), transposed convolution is performed similarly, while pooling and 1×1 convolutions are trivial, hence not discussed in further details.

## 4 EXPERIMENTAL RESULTS

### 4.1 EXPERIMENTAL SETUP

Our experiments are conducted on 4 Nvidia GPU platforms: *A100* (Ampere architecture), *H100* (Hopper), *Jetson* Orin Nano (Ampere), and RTX *2080 Ti* (Turing), all equipped with Tensor Cores. We evaluate models under the following execution settings: *PyTorch-FP32/FP16* – baseline implementations in PyTorch (Paszke et al., 2019) using 32-bit floating point and 16-bit floating point for both weights and activations, where the FP16 variant employs tensor cores; *MBU-Net* adopts binary activation and mixed binary/masked weights; The *Binary* model uses both binary activations and weights. The first and the last convolution layers in the quantized experiments are full-precision. For benchmarks, we use 3 representative datasets: *Carvana* (Shaler et al., 2017) – car image segmentation, *ISIC* (Codella et al., 2019) – skin lesion image segmentation, and *Nuclei* – nuclei segmentation in divergent images (Goodman et al., 2018). Dataset details are in Appendix A.4.

### 4.2 EFFICIENCY COMPARISON

We compare separate GPU and execution settings over the selected datasets. Efficiency is evaluated and discussed in terms of latency and energy in the following sections.

#### 4.2.1 LATENCY COMPARISON

Figure 5 shows the latency comparison. MBU-Net consistently outperforms FP32, with an average speedup of 4.83× over all platforms and datasets. On average, MBU-Net also outperforms FP16 with a 2.04× speedup. Noticeably, FP16 achieves better latency results, even compared to Binary. This is likely due to the removal of native BMMA support on Nvidia Hopper Tensor Core architecture (NVIDIA, 2022) – performance is lost, despite MBU-Net and Binary models still being runnable. INT8 and INT4 results are also presented for A100 using TensorRT (NVIDIA, 2025). Although INT8 achieves lower latency than FP16, it is outperformed by MBU-Net by 1.52×. However, INT4 shows inferior performance compared to INT8 likely due to the overhead in packing/unpacking. Consequently, MBU-Net shows 2.23× speedup versus INT4.

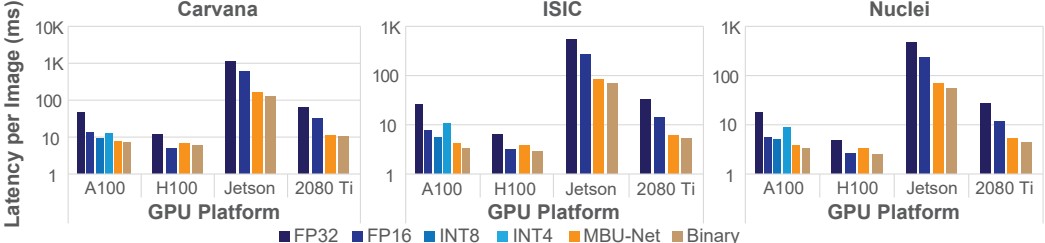

Figure 5: Latency per image inference (milliseconds). Lower is better.

#### 4.2.2 ENERGY COMPARISON

Figure 6 presents the energy comparison. Consistently, MBU-Net's energy cost is comparable to the Binary model, while achieving average energy reduction of 8.53× over FP32 and 3.54× over FP16. On A100, it achieves 2.09× and 2.63× improvements over INT8 and INT4, separately.

### 4.3 ACCURACY COMPARISON

Table 2 compares the U-Net model across different precision configurations. "Full Precision" refers to both full-precision weights and activations (FP32 and FP16 show negligible accuracy differences), while other configurations employ binary activations. Accuracy is evaluated with three metrics: Dice Score, IoU, and F1 score, with our results highlighted in the shaded row. On average, aggressive quantization in MBU-Net only leads to drops of 0.029, 0.037, and 0.024 in Dice, IoU, and F1 scores.

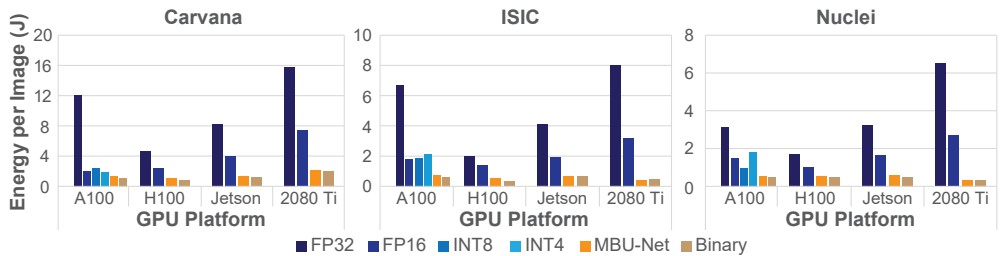

Figure 6: Energy per image inference (Joule). Lower is better.

Table 2: Accuracy comparison using Dice score, IoU, and F1 score across three datasets. All metrics range from 0 to 1, higher is better. All configurations other than Full Precision use binary activations.

| Dataset | Carvana | | | ISIC | | | Nuclei | | |
|---|---|---|---|---|---|---|---|---|---|
| Metric (Weight-Activation) | Dice | IoU | F1 | Dice | IoU | F1 | Dice | IoU | F1 |
| Full Precision[*] | 0.997 | 0.994 | 0.997 | 0.771 | 0.644 | 0.783 | 0.867 | 0.776 | 0.874 |
| INT8 | 0.994 | 0.987 | 0.994 | 0.763 | 0.633 | 0.776 | 0.823 | 0.741 | 0.851 |
| INT4 | 0.989 | 0.979 | 0.989 | 0.753 | 0.619 | 0.765 | 0.819 | 0.739 | 0.850 |
| MBU-Net | 0.981 | 0.963 | 0.981 | 0.750 | 0.617 | 0.763 | 0.817 | 0.722 | 0.839 |
| Binary | 0.662 | 0.530 | 0.693 | 0.560 | 0.399 | 0.570 | 0.434 | 0.302 | 0.464 |

[*] Both weights and activations are in full precision; negligible difference between FP32 and FP16 accuracy.

## 4.4 TRADE-OFF ANALYSIS

Figure 7 presents the Pareto frontier illustrating the trade-off between accuracy (Dice score) and speed (FPS) across different MBU-Net configurations on A100 GPU, where masked layers are chosen using the cost-aware masking strategy with $w_{op} = w_{param} = 0.5$. The corresponding cost score ranking and additional results with other metrics are in Appendix A.5.

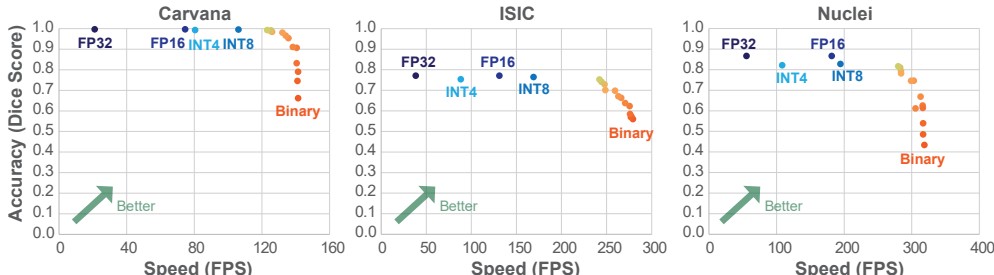

Figure 7: Pareto frontier of accuracy (dice score) and speed (FPS) on A100 for 3 datasets. Colors (yellow →red) indicate more layers get masked, based on the cost-aware masking strategy in §3.1.2.

## 5 CONCLUSION

In this work, we address the challenges of bringing binary quantization to U-Net for real-time image segmentation on edge devices. Two critical observations are identified: (1) The necessity of an explicit zero state to suppress noisy signals and promote sparsity, and (2) the uniform sensitivity of U-Net layers to quantization. Building on these insights, we propose a cost-aware masking strategy that balances accuracy and efficiency, resulting in Masked Binary U-Net (MBU-Net). To realize its benefits on commodity hardware, we develop a GPU execution framework that leverages Tensor Cores through a subtractive bit-encoding scheme, efficiently supporting masked binary weights with binary activations. This framework ensures high throughput and power efficiency, while remaining deployable on widely available edge GPUs.

Experiments across multiple segmentation datasets demonstrate that MBU-Net achieves near full-precision accuracy (3% drop on average), while attaining 2.04× speedup and 3.54× lower energy consumption compared to a 16-bit floating point U-Net. These results establish MBU-Net as a practical and scalable solution for real-time, high-resolution image segmentation on Tensor Cores. If deployed on AR/VR, autonomous drones or related domains, this can raise a 30 FPS pipeline to 60 FPS, and cut segmentation power by 70%, extending battery life. While measured on GPU Tensor Cores, MBU-Net is well suited for ASICs: its low-bit width arithmetic and masking enable narrow data paths and reduced SRAM bandwidth, yielding higher performance per watt and deterministic latency in a dedicated accelerator.

ACKNOWLEDGMENTS

This work is supported by the U.S. Department of Energy, Office of Science, Office of Advanced Scientific Computing Research, for the DeCoDe project at Pacific Northwest National Laboratory, in support of the MEERCAT Microelectronics Science Research Center. This work is also supported by the U.S. Department of Energy, Office of Electricity, through its Advanced Grid Modeling (AGM) program, by DARPA under Contract W912CG25CA007, and by NSF under Award No. 2326494. This research used resources of the National Energy Research Scientific Computing Center (NERSC), a U.S. Department of Energy Office of Science User Facility at Lawrence Berkeley National Laboratory, under Contract No. DE-AC02-05CH11231 using NERSC award NERSC DDR-ERCAP0035256. The Pacific Northwest National Laboratory is operated by Battelle for the U.S. Department of Energy under Contract DE-AC05-76RL01830. Finally, we would like to thank the anonymous reviewers for their feedback, and META for the collaboration.

REPRODUCIBILITY STATEMENT

All datasets employed in our experiments are publicly available and described in detail in Appendix Section A.4.2. The experimental settings are presented in Section 4.1 and further elaborated in Appendix Section A.4. The code is accessible at https://github.com/ChunshuWu/MBU-Net.

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

# A APPENDIX

## A.1 LLM USAGE

We use GPT-5, Gemini-2.5-Pro, and Claude-4-sonnet to polish writing and assist literature searching. All LLM-assisted text and references have been reviewed, revised, and verified by the authors.

## A.2 DETAILED MBU-NET ARCHITECTURE

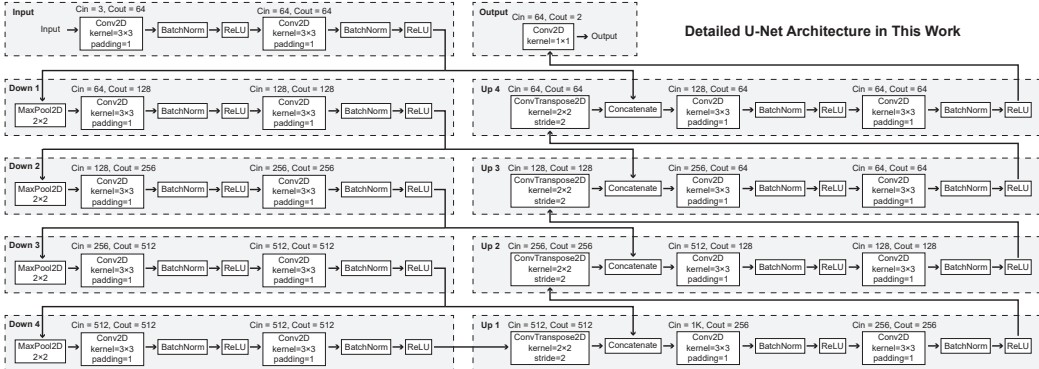

Figure A.1: The foundational U-Net architecture used in this work.

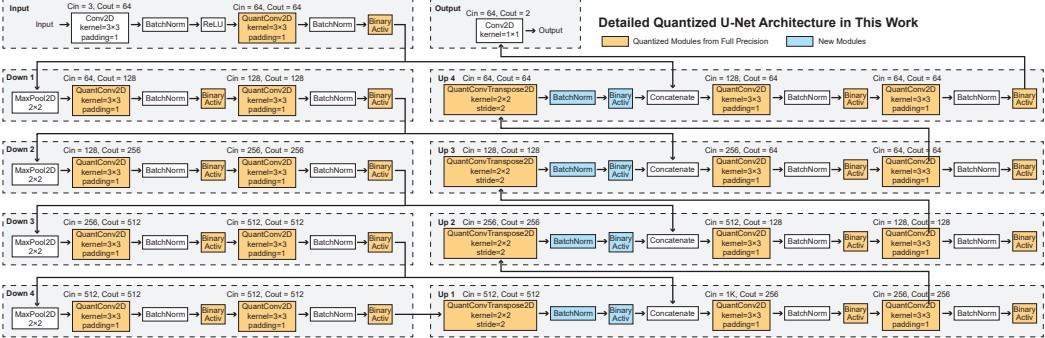

Figure A.2: The quantized U-Net architecture used in this work.

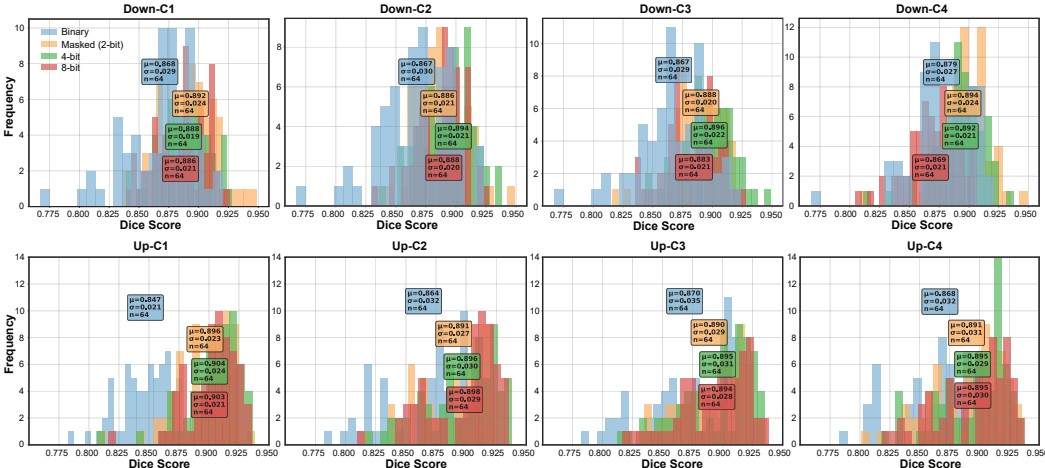

Figure A.3: Per-layer quantization space exploration. First row: Dice score distribution of 256 experiments on down-sampling convolution layers with binary remaining layers. Second row: distribution of 256 experiments on up-sampling convolution layers with binary remaining layers. Each subfigure explores the accuracy improvement of increasing the number of bits for this layer's weights.

Figure A.1 and Figure A.2 illustrate the model architectures used in this work, with the former used to evaluate the full-precision configuration, and the other for quantized experiments. Compared to the foundational U-Net architecture, the quantized architecture not only modifies the ReLU activation function and the convolution/transposed convolution into binary versions, but also adds batchnorm and binary activation after transposed convolution layers to preserve the binary activation and ensure training quality.

## A.3 QUANTIZATION SPACE EXPLORATION

Figure A.3 shows the statistics of using higher bit quantization for individual convolution layers using Carvana dataset. The two input convolution layers, and the final convolution layer are full-precision (slightly different from the experiments in the main text, where the 2nd convolution layer is always masked-binary). In the first row, the four down-sampling layers are traversed from all binary to all 8-bit, totaling $4^4 = 256$ experiments, while the remaining layers are binary. Similarly, the second row traverses all double convolution layers in up-sampling with another 256 experiments. The distribution demonstrates that for most of the layers, the transition from binary to masked-binary contributes the most to accuracy, whereas negligible benefit is obtained from 2-bit to 4-bit, or from 4-bit to 8-bit. We thus confine the paper's scope within binary and 2-bit.

## A.4 TRAINING AND INFERENCE CONFIGURATIONS

### A.4.1 HYPERPARAMETERS

Training and inference parameters are shown in Tables A.1 and A.2 list the hyperparameters. Full-precision training uses the RMSprop optimizer, while quantized training uses Adam.

Table A.1: The training hyperparameters for separate experiments.

| Batch Size | Learning Rate | Number of Epochs | Grad Clipping | Weight Decay | Momentum |
|------------|---------------|------------------|---------------|--------------|----------|
| 2 | 1e-5 | 30 | 1.0 | 1e-8 | 0.9 |

Table A.2: The batch sizes used for inference experiments. For fair comparison, the batch sizes for FP32 and FP16 are maximized before the GPUs run out of memory.

| Dataset | Carvana | | | ISIC | | | Nuclei | | |
|---------|------|------|---------|------|------|---------|------|------|---------|
| Configuration | FP32 | FP16 | MBU-Net | FP32 | FP16 | MBU-Net | FP32 | FP16 | MBU-Net |
| A100 | 32 | 32 | 64 | 64 | 128 | 128 | 64 | 128 | 128 |
| H100 | 64 | 64 | 64 | 128 | 256 | 256 | 128 | 256 | 256 |
| Jetson | 4 | 8 | 32 | 8 | 16 | 64 | 8 | 16 | 64 |
| 2080 Ti | 8 | 16 | 64 | 16 | 32 | 128 | 16 | 32 | 128 |

### A.4.2 DATASET DETAILS

Table A.3 shows the dimensions of the three selected datasets. The data are scaled from the original datasets to the numbers listed in the table. For Nuclei dataset, the original individual nucleus masks are combined into a comprehensive mask to facilitate the image-mask pair for segmentation.

Table A.3: Dataset specifications.

| Dataset | Number of Images | Width | Height |
|---------|------------------|-------|--------|
| Carvana | 5088 | 959 | 640 |
| ISIC | 3693 | 640 | 480 |
| Nuclei | 669 | 512 | 512 |

## A.5 DETAILED ABLATION STUDIES

### A.5.1 COST RANKING

The cost scores and rankings for individual layers of the U-Net used in this work are listed in Table A.4. In our ablation studies, the layers are gradually masked starting from Rank 1 to evaluate the trade-off between efficiency and accuracy.

Table A.4: Cost scores and rankings of layers.

| Rank | Layer | Cost Score | Rank | Layer | Cost Score | Rank | Layer | Cost Score |
|------|-------|------------|------|-------|------------|------|-------|------------|
| 1 | Up-CT4 | 0.011 | 5 | Up-C3 | 0.228 | 9 | Up-C4 | 0.512 |
| 2 | Up-CT3 | 0.016 | 6 | Down-C1 | 0.273 | 10 | Up-C1 | 0.521 |
| 3 | Up-CT2 | 0.037 | 7 | Up-C2 | 0.286 | 11 | Down-C4 | 0.583 |
| 4 | Up-CT1 | 0.120 | 8 | Down-C2 | 0.406 | 12 | Down-C3 | 0.625 |

### A.5.2 RESULTS OF ADDITIONAL METRICS

Figure A.4 shows the Pareto frontiers of speed in FPS and accuracy in IoU and F1 score, on A100 GPU platform. The results are consistent with Figure 7 in the main text. Figure A.5 demonstrates the Pareto frontiers on other GPU platforms (H100, RTX 2080 Ti, and Jetson Orin Nano). On RTX 2080 Ti and Jetson Orin Nano, MBU-Net variations clearly outperforms FP32 and FP16 in terms of speed, while the accuracy drop is low when sufficient masked layers are deployed. As discussed in the main text, on H100, the performance of MBU-Net is lost due to the lack of BMMA support.

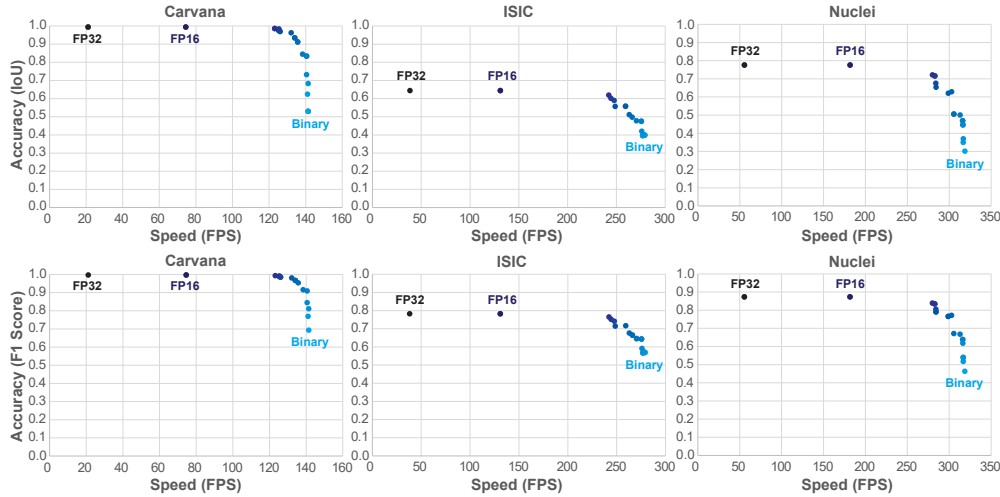

Figure A.4: Speed vs. accuracy (IoU and F1 score) on A100.

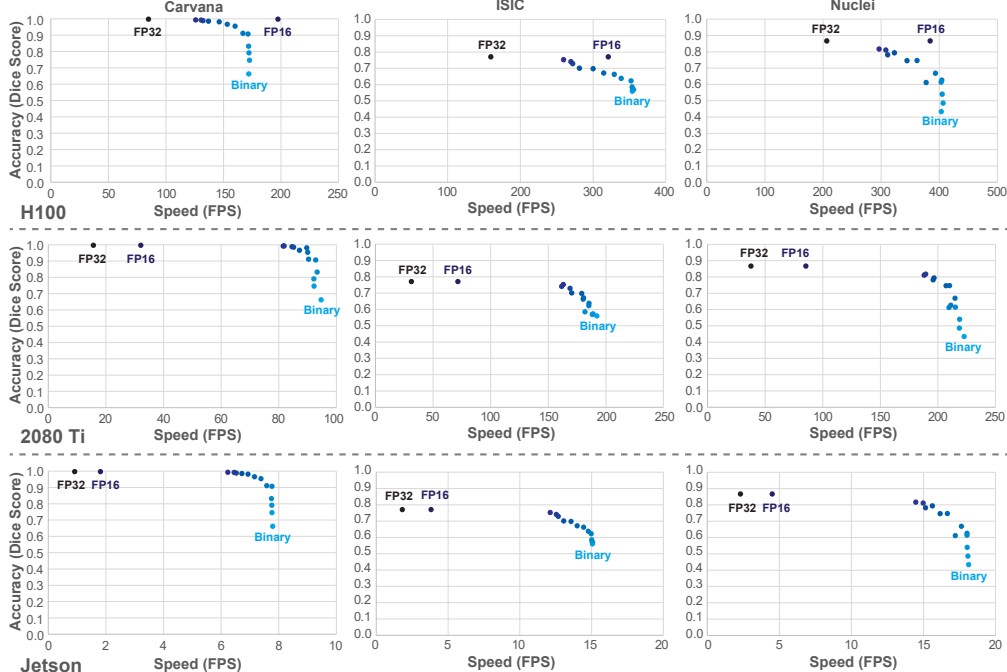

Figure A.5: Speed vs. accuracy (Dice score) on H100, RTX 2080 Ti, and Jetson Orin Nano.

