# OpenReview forum: "Zeros can be Informative: Masked Binary U-Net for Image Segmentation on Tensor Cores"
_ICLR.cc/2026/Conference — ICLR 2026 Poster_

### Official Review · Reviewer_pWxq · 2025-10-30

**Soundness:** 2
**Presentation:** 2
**Contribution:** 3
**Rating:** 2
**Confidence:** 4

**Summary:**

To address the computational and power limitations of U-Net for high-resolution image segmentation on edge devices, the authors propose a Masked Binary U-Net (MBU-Net) method. Through the experimental analysis of binary U-Net, the authors found that introducing zero mask training produces significant sparsity (more than 95% of the layers have zero weights), and the sensitivity of the layers to the mask is roughly uniform. Motivated by this observation, the authors design a cost-aware masking strategy with three-valued weights (-1, 0, +1) and binary activations, and develop an efficient inference framework based on GPU Tensor Core. Experimental results on three medical/image segmentation datasets show that the proposed method achieves about 2x speedup and 3.54x energy reduction compared to FP16 U-Net, but at the cost of 3% accuracy degradation on average. The authors note that the method relies on the native binary matrix multiplication instructions of the GPU, which is limited in performance on the H100 architecture with this support removed, and the accuracy loss caused by binarization may not be acceptable in some application scenarios.

**Strengths:**

1.In this paper, the zero masking state in the weights of a binary U-Net improves the segmentation quality.

2.The authors report latency evaluations across multiple GPU platforms, demonstrating an average 2.04× speedup and 3.54× power efficiency improvement compared to a 16-bit floating-point U-Net baseline.

**Weaknesses:**

1.The authors only evaluated their method on the typical U-Net architecture shown in Fig. 2, without providing results on other segmentation models, which undermines the robustness and generalizability of the proposed approach to other network architectures.

2.The BACKGROUND and RELATED WORK sections should be merged into a single section.

**Questions:**

1.The authors conducted tests on A100, H100, RTX 2080Ti, and Jetson Orin Nano. However, it is unclear how the different hardware architectures of these GPUs affect the proposed method. For instance, A100 uses NVIDIA Ampere architecture, H100 uses Hopper architecture, and RTX 2080Ti uses Turing architecture. What are the implications of these architectural differences on the performance and applicability of the proposed approach?

2.The authors propose a zero masking method in the manuscript, but the logic and parameter settings for applying zero masking to weights are not clearly explained. Is this an adaptive adjustment or a fixed threshold approach?

3.The authors propose a subtractive bit-encoding method tailored for Tensor Cores in commercial GPUs. However, can this method be generalized into a universal and elegant hardware design framework or dataflow scheduling strategy that provides insights for hardware platforms beyond GPUs, rather than serving merely as an experimental optimization technique specific to GPUs?

4.Please see the Weaknesses section for other issues.

---

> ### Author Response · Authors · 2025-12-02
> **Response to Reviewer pWxq**
>
> We thank the reviewer for the thoughtful feedback and suggestions.
>
> **W1. Generalizability.** We confirm that our method is inherently generalizable because it is implemented via generic matrix multiplication operations. Since MBU-Net realizes the zero state by formulating it as a dense matrix multiplication (Equation 3), it is mathematically compatible with any architecture relying on standard convolution or linear layers. We deliberately confined our scope to the foundational U-Net architecture to establish a rigorous baseline for this quantization paradigm. U-Net allows us to demonstrate a critical efficiency phenomenon: although introducing a zero state approximately doubles the operation cost of a layer, applying this to cheap layers like Transposed Convolutions contributes to negligible computational overhead. In contrast, the gain in accuracy is massive, effectively maximizing the performance-to-cost ratio. Extending this method to more complex variants introduces additional architectural variables that require deeper investigation outside the scope of this paper.
>
> **W2. Merging of background and related work.** We accept this suggestion and have merged the related work into background.
>
> **Q1. Impact of different hardware architectures.** The performance variation stems from the availability of native binary matrix instructions. The NVIDIA Turing (RTX 2080Ti) and Ampere (A100, Jetson Orin) architectures provide native support for Binary Matrix Multiply-Accumulate (BMMA) instructions, allowing MBU-Net to fully leverage hardware acceleration for 1-bit inputs. In contrast, the Hopper architecture (H100) removed native BMMA support, requiring emulation that reduces efficiency. This distinction is crucial for our target domain, as edge-class devices (like the Jetson Orin evaluated in our work) continue to support these efficient binary instructions.
>
> **Q2. The logic for zero masking.** We clarify that the parameters governing the masking logic are fixed hyperparameters established prior to training, rather than adaptively tuned values. In our main experiments, we utilized a balanced setting to determine the cost score for each layer. These hyperparameters are used to generate a static priority ranking of layers based on their hardware cost.
>
> **Q3. Generalization of subtractive bit-encoding.** We confirm that this method is applicable beyond GPUs. The core principle of subtractive bit-encoding is decomposing a complex arithmetic operation (ternary MAC) into fundamental bit-wise logic (XOR and Popcount). This is a universal design pattern in digital logic design, aligning with bit-serial processing techniques. Therefore, this dataflow strategy is highly portable to other hardware platforms, such as FPGAs or custom ASICs, where bit-wise operations are primitive and can be implemented with even greater efficiency than on general-purpose GPUs.

---

### Official Review · Reviewer_MDEM · 2025-10-31

**Soundness:** 2
**Presentation:** 3
**Contribution:** 3
**Rating:** 6
**Confidence:** 4

**Summary:**

This paper proposes a masked-binary mechanism with cost-aware layer selection that adds a "zero state" to binary networks, improving expressiveness and stability while preserving bit-operator efficiency. Masking selective low-cost layers achieves near "full-mask" accuracy with predictable hardware performance.

**Strengths:**

1. The paper addresses real-time, high-resolution segmentation on edge devices, focusing on accuracy, latency, and energy. The empirical insights are straightforward and motivate a practical design, making the contribution both coherent and relevant.
2. The work combines masked binary weights with a cost-aware layer selection and a GPU execution framework using Tensor Cores. The subtractive bit-encoding and native binary operations show strong engineering rigor and enable deployable efficiency gains.
3. Experiments across multiple GPUs and datasets demonstrate near full-precision accuracy with notable speed and energy improvements. The comprehensive metrics and ablations support robustness, portability, and practical relevance.

**Weaknesses:**

1. The method adds ternary weights to selected layers of a binary U-Net to approach full-precision accuracy with near-binary efficiency. However, the paper does not clearly compare against ternary quantization baselines. Could the authors clarify in which dimensions MBU-Net outperforms classical ternary methods?
2. The paper enhances binary networks by masking some weights to zero. Intuitively, This seems related to sparsity/pruning. Could the authors clarify: can this method be considered as a combination of pruning and binary quantization? If not, what are the key distinctions?
3. While the analysis and experiments are convincing, the authors focus on the classical U-Net. I am concerned about robustness and generalizability: can the proposed method be generalized to similar segmentation architectures (e.g., U-Net variants) for broader applicability?

**Questions:**

Please see Weaknesses.

---

> ### Author Response · Authors · 2025-12-02
> **Response to Reviewer MDEM**
>
> We thank the reviewer for the positive assessment and for recognizing the engineering rigor and practical value of our work.
>
> **W1. Comparison against ternary baselines.** We clarify that our experiments in Figure 7 already include a fully ternary configuration, represented by the data point where *all* layers have zero states. Our approach advances beyond classical ternary methods in two ways. First, while classical methods typically enforce uniform ternary quantization, our cost-aware strategy enables a flexible design space that selectively masks critical layers rather than masking the entire network. Second, while prior ternary works primarily remained at the algorithmic level or focused on custom hardware, we provide a concrete end-to-end deployment framework on general-purpose GPU Tensor Cores.
>
> **W2. Relationship to pruning.** MBU-Net is fundamentally a quantization strategy rather than a pruning or sparsity-oriented technique. However, it is worth mentioning that MBU-Net enables sparsity. As noted in our empirical observations (Finding 1), our masking strategy results in approximately 90% of the weights in masked layers being zero. While our current implementation treats these zeros mathematically within a dense matrix multiplication framework, this high degree of induced sparsity presents a significant opportunity for future research.
>
> **W3. Generalizability.** We confirm that our method is applicable to other architectures because it is implemented via generic matrix multiplication operations. Since MBU-Net realizes the zero state by formulating it as a dense matrix multiplication (Equation 3), it is mathematically compatible with any architecture relying on standard convolution or linear layers. However, we deliberately confine our scope to the foundational U-Net architecture to establish a rigorous baseline for this new quantization paradigm. Extending this method to more complex variants introduces additional architectural variables that require deeper investigation outside the scope of this paper. We therefore position these extensions as promising directions for future work.

---

### Official Review · Reviewer_UTpC · 2025-10-31

**Soundness:** 2
**Presentation:** 3
**Contribution:** 2
**Rating:** 2
**Confidence:** 3

**Summary:**

The author’s propose a real-time image segmentation approach using UNet which they name Masked Binary UNet. The approach includes an explicit zero-state (which was found to be essential for performance), and layers are quantized uniformly across layers along with an execution framework that maps the model to TensorCores. They show that this approach performs much faster than the full precision variants while maintaining most of its original performance.

**Strengths:**

S1 - Interesting finding that including zero-mask values seems to preserve the performance of the UNet model. Moreover, an interesting tensor-core deployment scheme was shown.

S2 - The proposed system performs roughly just as well as the full-precision UNet’s while being much faster.

**Weaknesses:**

W1 - While INT8 and INT4 models are evaluated for performance (in 4.3), their latency / speed / energy is not shown. Could it be that INT8 and INT4 perform just as well as the proposed method in terms of speed?

W2- Authors should’ve compared to another simple baseline, namely using TensorRT for inference optimization / quantization as in https://arxiv.org/pdf/2012.12259.

W3 - The paper is lacking in experimental results. For example, Section 4.4 summarizes insights already gathered in 4.3 and 4.2.1. On the whole, the table in 4.3 could’ve just had an extra column indicating the FPS and nothing would’ve been lost if 4.2.1 and 4.4 were removed.

**Questions:**

* Q1 - Why is Section 4.4 considered an ablation study? No parts of the system were ablated? Could’ve tried random costs on the layers as opposed to the weighting scheme as well.
* My main concern is W1, followed by W2 and W3. Namely, why didn't the authors show INT8 and INT4 in the pareto curve?

---

> ### Author Response · Authors · 2025-12-02
> **Response to Reviewer UTpC**
>
> **W1. Performance of INT8 and INT4.** We thank the reviewer for this valuable suggestion. In the revised manuscript, we have integrated TensorRT-optimized INT8 and INT4 baselines into the evaluation.
>
> *Latency & Energy (Figures 5 & 6).* We explicitly added INT8 and INT4 comparisons on the A100 platform. The results demonstrate that MBU-Net achieves lower latency and energy consumption compared to INT8. Notably, we observed that INT4 inference is slower than INT8 due to the lack of native INT4 Tensor Core support, which involves overhead-heavy operations. This highlights the hardware-efficiency advantage of our binary approach, which is natively supported on the platform.
>
> *Pareto Frontier (Figure 7).* We have updated the Pareto curves to include these quantization baselines, providing a clear visual comparison of the accuracy-efficiency trade-off.
>
> **W2. Comparison to TensorRT baselines.** We agree that a standard industry baseline is essential. The INT8 and INT4 models introduced in response to W1 serve this purpose. These baselines are implemented using NVIDIA’s standard TensorRT optimization pipeline.
>
> **W3. Organization of experimental results.** Despite some redundancy, we respectfully maintain Sections 4.2 and 4.4, as they serve critical purposes that a single table cannot fully capture:
> *Hardware Generality (Section 4.2).* This section is necessary to demonstrate that our gains are not specific to a single GPU architecture but generalize across varying hardware constraints. A single table is insufficient to demonstrate these cross-platform scaling behaviors.
> *Design Space Exploration (Section 4.4).* The Pareto Frontier (Figure 7) visualizes the trade-off between accuracy and speed, a level of details not covered in the table. This is crucial for demonstrating the flexibility of our strategy, allowing practitioners to navigate the design space.
>
> **Q1. Ablation Study.** Our ablation refers to gradually removing zero states in the layers. To avoid ambiguity, we have renamed the section as “Trade-Off Analysis”.

---

### Official Review · Reviewer_P1M6 · 2025-11-01

**Soundness:** 3
**Presentation:** 3
**Contribution:** 3
**Rating:** 6
**Confidence:** 3

**Summary:**

In this paper, the author argues that traditional binary U-Nets lose too much accuracy and lack efficient GPU implementations. To address this, the paper proposes MBU-Net, which introduces a zero state to make weights ternary and applies it selectively through a cost-aware masking strategy. It also builds a custom GPU framework using subtractive bit encoding so ternary weights can run efficiently on Tensor Cores, achieving near full-precision accuracy with much faster and more energy-efficient inference.

**Strengths:**

* This paper is well organized and easy to follow, especially for readers who are not familiar with this area.
* The insights that introducing large amount of ‘zero state’ into pure binary U-Nets is extremely helpful on segmentation task is great
* A major strength of this work is its practical GPU execution framework, which provides tangible, measurable speedup on widely available NVIDIA GPUs
* Innovatively unlocks Hardware Potential with “Subtractive Bit-Encoding”, extending the BMMA (Binary matrix multiply) to the ternary computation.

**Weaknesses:**

* Introducing ‘zero state’ into pure binary U-Net can significantly boost performance on segmentation, the performance of such a method applied on other types of networks and tasks remains unclear. However, the paper’s innovation on the low-level hardware implementation is still very solid.

**Questions:**

Please refer to the weakness part

---

> ### Author Response · Authors · 2025-12-02
> **Response to Reviewer P1M6**
>
> We thank the reviewer for the positive assessment and for recognizing the value of our work.
>
> This method is generalizable to other model architectures for two key reasons. First, in terms of effectiveness, prior works in Related Work (now Section 2.3) have consistently demonstrated that ternary networks improve accuracy over binary networks across various domains. Second, in terms of feasibility, our method is implemented via standard matrix multiplication, a generic operation fundamental to almost all deep learning models.
>
> However, we confine the paper’s scope on U-Net because it represents a unique and critical case of computational imbalance. In U-Net, Transposed Convolution layers are computationally lightweight yet highly sensitive to information loss. This allows us to highlight the efficiency of adding a zero state: although introducing a zero state approximately doubles the operation cost, applying this to cheap layers like Transposed Convolutions contributes to negligible computational overhead. In contrast, the gain in accuracy is massive, effectively maximizing the performance-to-cost ratio. We view this observation as a foundational step for a broader research direction. We hypothesize that (1) such “cheap yet critical” layers are prevalent in modern models and (2) injecting lightweight zero-masked layers into existing binary networks can effectively improve accuracy in these general cases.

---

### Meta-Review · Area_Chair_RMTr · 2025-12-11

**Summary:**

This paper proposes, a masked-binary U-Net architecture that introduces a “zero state’’ to ternarise binary weights and employs a cost-aware masking strategy to selectively apply ternary weights to low-cost layers. The authors further develop a custom GPU execution framework based on subtractive bit-encoding, enabling ternary operations to be mapped efficiently to Tensor Cores and thereby achieving substantial speed and energy gains with accuracy close to the full-precision U-Net.

Across reviewers, the paper was seen as well-written, easy to follow, and a practical solution for deploying real-time segmentation on edge devices. Reviewers also agreed that the hardware execution framework is a significant engineering contribution, demonstrating meaningful improvements in latency and energy across multiple GPUs. The empirical finding that injecting a large fraction of zero weights into binary U-Net improves segmentation accuracy is viewed as interesting and insightful.

Reviewers also raised substantial concerns, particularly about the experimental completeness, baseline comparisons, and generalizability. The provided experiments and concise rebuttal have effectively resolved these concerns. Therefore, this paper is recommended for acceptance.

**Reviewer Concerns:**

UTpC.

Could it be that INT8 and INT4 perform just as well as the proposed method in terms of speed?  Results added and problem solved.

Authors should’ve compared to tensorRT for inference optimization. Already compared.

MDEM : paper does not compare against classical ternary quantization methods. Already compared.


pWxq: it is unclear how the different hardware architectures of these GPUs affect the proposed method. Solved.

**Reviewer Scores:**

P1M6  maintain to 6.
UTpC increases to 6 from 2.

MDEM maintains 6.

pWxq  increases to 6 from 2.

---

### Decision · Program_Chairs · 2026-01-26

Accept (Poster)